# Tumor Necrosis Factor (TNF)-α-Stimulated Gene 6 (TSG-6): A Promising Immunomodulatory Target in Acute Neurodegenerative Diseases

**DOI:** 10.3390/ijms24021162

**Published:** 2023-01-06

**Authors:** Daniele La Russa, Chiara Di Santo, Ignacio Lizasoain, Ana Moraga, Giacinto Bagetta, Diana Amantea

**Affiliations:** 1Section of Preclinical and Translational Pharmacology, Department of Pharmacy, Health and Nutritional Sciences, University of Calabria, 87036 Rende, CS, Italy; 2Unidad de Investigación Neurovascular, Departamento de Farmacología y Toxicología, Facultad de Medicina, Instituto Universitario de Investigación en Neuroquímica, Universidad Complutense de Madrid, and Instituto de Investigación Hospital 12 de Octubre (Imas12), 28040 Madrid, Spain

**Keywords:** neurodegeneration, neuroinflammation, stroke, TSG-6

## Abstract

Tumor necrosis factor (TNF)-α-stimulated gene 6 (TSG-6), the first soluble chemokine-binding protein to be identified in mammals, inhibits chemotaxis and transendothelial migration of neutrophils and attenuates the inflammatory response of dendritic cells, macrophages, monocytes, and T cells. This immunoregulatory protein is a pivotal mediator of the therapeutic efficacy of mesenchymal stem/stromal cells (MSC) in diverse pathological conditions, including neuroinflammation. However, TSG-6 is also constitutively expressed in some tissues, such as the brain and spinal cord, and is generally upregulated in response to inflammation in monocytes/macrophages, dendritic cells, astrocytes, vascular smooth muscle cells and fibroblasts. Due to its ability to modulate sterile inflammation, TSG-6 exerts protective effects in diverse degenerative and inflammatory diseases, including brain disorders. Emerging evidence provides insights into the potential use of TSG-6 as a peripheral diagnostic and/or prognostic biomarker, especially in the context of ischemic stroke, whereby the pathobiological relevance of this protein has also been demonstrated in patients. Thus, in this review, we will discuss the most recent data on the involvement of TSG-6 in neurodegenerative diseases, particularly focusing on relevant anti-inflammatory and immunomodulatory functions. Furthermore, we will examine evidence suggesting novel therapeutic opportunities that can be afforded by modulating TSG-6-related pathways in neuropathological contexts and, most notably, in stroke.

## 1. Introduction

Stroke is a leading cause of mortality and disability worldwide, and implementation of prevention and treatment strategies is imperative to limit the expected growth of its global burden [1]. To date, the only available therapeutic strategy for acute ischemic stroke consists in thrombus lysis/removal achieved by intravenous administration of human recombinant tissue plasminogen activator (rt-PA) or other proteins with similar activity [2,3,4] and/or endovascular thrombectomy or embolectomy [5,6,7,8]. These interventions are based on reperfusion of the infarcted area; while they do not exert direct neuroprotection to block the progression of brain damage, that remains a hopeful target. To this end, in the last several decades, substantial progress has been made to delineate the complex molecular processes underlying the pathobiological mechanisms implicated in ischemic brain damage [9,10]. Decreased blood flow to the brain triggers an ischemic detrimental cascade involving excitotoxicity, oxidative stress, and inflammation that contribute to the development of cerebral damage [11,12,13]. Moreover, several studies support the evidence that central and peripheral immune responses play a pivotal role in ischemic pathobiology, exerting detrimental or protective functions, depending on the specific molecules or cells (phenotypes/subtypes) involved, as well as on the phases after the initial insult [14,15,16]. To implement therapeutic approaches, recent experimental work has suggested that an appropriate immunomodulation would allow to achieve neuroprotection. Thus, dissecting the molecular mechanisms by which the immune system is implicated in ischemic stroke injury is crucial for the identification of novel therapeutic targets. To this end, in this review, we will discuss the most recent data on the involvement of tumor necrosis factor (TNF)-α-stimulated gene 6 (TSG-6) in acute and chronic neurodegenerative diseases, particularly focusing on relevant anti-inflammatory and immunomodulatory functions. Furthermore, we will examine evidence suggesting novel therapeutic opportunities that can be afforded by modulating TSG-6-related pathways in neuropathological contexts and, most notably, in stroke.

## 2. Inflammatory and Immune Mechanisms in Ischemic Stroke

Damage-associated molecular pattern molecules (DAMPs), released following ischemic cell death, trigger inflammatory responses by interacting with pattern recognition receptors, including nucleotide oligomerization domain (NOD)-like receptors, receptor of advanced glycation end-products (RAGE), and Toll-like receptors (TLRs), the latter being involved in the initiation of innate immunity [17,18,19]. Upon ligand binding, TLR4 dimerizes and promotes myeloid differentiation primary response (MyD)88- or TIR-domain-containing adapter-inducing interferon-β (TRIF)-dependent signaling, implicated in downstream activation of nuclear factor kappa B (NF-κB) that is, in turn, necessary for inflammatory cytokine production [20,21].

After sensing tissue damage, resting ramified microglia gradually transform into amoeboid-shaped cells that initially clear debris and foster tissue repair [22,23,24]. These ‘beneficial’ functions, typically provided by M2-like phenotypes, are rapidly replaced by a transition toward pathological M1 subsets [25] that express a range of inflammatory mediators, such as TNF-α, interleukin (IL)-1β, IL-6, inducible nitric oxide synthase (iNOS), monocyte chemoattractant protein (MCP)-1, and macrophage inflammatory protein (MIP)-1α [26,27,28,29,30]. Indeed, a range of diverse transcriptional programs have been reported to occur in rodent and human microglia, since, in the late stages, M2-like phenotypes release proresolving cytokines (i.e., IL-10 and transforming growth factor-β, TGF-β) and trophic factors that exert proangiogenic and reparative functions [31,32,33,34,35].

Stroke-induced sterile inflammation also involves mobilization and cerebral recruitment of circulating monocytes, neutrophils, and T cells [36,37,38,39,40]. Regarding monocytes, classical CD14+/CD16- subtypes mainly secrete inflammatory TNF-α, IL-6 and IL-1β [41,42,43], and their blood number, as well as their expression of TLR-4, increase early after stroke, being independently associated with poor outcome in patients [44,45,46]. Moreover, higher monocyte counts positively correlate with ischemic stroke severity and adverse prognosis [47,48,49], whereas the less represented CD16+ intermediate and nonclassical monocytes have been found to be inversely related to poor outcomes and mortality, respectively [44], likely exerting protective/reparative functions [50]. Once recruited in the injured tissue, bloodborne monocytes transform into subsets of macrophages, as shown both in patients and in rodent models [51,52,53].

In the early stage after ischemia, alternatively-activated M2 subsets predominate in the ischemic core region, where they display reparative functions [54]. Thereafter, they switch toward M1-like phenotypes that abound in the injured brain to elicit detrimental effects through the release of inflammatory and toxic mediators, including TNF-α, IL-1β, and ROS [25,29,35,40,55,56,57].

The dualistic role of innate immunity in ischemic pathophysiology is further underscored by ability of neutrophils to adopt N1 inflammatory or N2 beneficial phenotypes by TLR4-dependent mechanisms among other mechanisms [58,59,60,61]. Early after injury, they release inflammatory and neurotoxic mediators, including cytokines, ROS and proteases that prompt BBB damage, tissue edema, hemorrhagic transformation, and cerebral damage [59,62]. Once recruited to the ischemic tissue, neutrophils contribute to thrombus formation and expansion and impair revascularization and vascular remodeling through the release of neutrophil extracellular traps (NETs) [63,64]. Indeed, increased blood or brain-infiltrating neutrophils are associated with stroke severity and poor functional outcome [65]. However, the attempts to block these detrimental immune cells resulted in limited therapeutic success [59,65], likely due to the beneficial roles of N2 phenotypes that facilitate macrophage phagocytosis and may support neuroprotection [66,67,68].

A recent bioinformatic study revealed that among key immunoregulatory pathways implicated in ischemic stroke pathogenesis, TNF-related signaling covers a pivotal role [69]. TNF is an inflammatory cytokine with dualistic effects in the brain [70,71], since its soluble form is detrimental, whereas its membrane-bound form displays protective properties and is required for the maintenance of innate immunity [72,73,74,75,76]. After either experimental or human stroke, cerebral expression of TNF and of its receptors, TNFR1 and TNFR2, increases [77,78,79,80,81], whereby microglia and infiltrating macrophages represent the major source of TNF-α in the acute phase after ischemia [80,82]. Intriguingly, TNF-α mRNA levels are elevated in circulating monocytes isolated from ischemic stroke patients 24–48 h after symptoms onset [83], and both TNFR1 and TNFR2 are differentially regulated in distinct subpopulations of monocytes and neutrophils [84].

## 3. Biological Effects of TSG-6

Among the multitude of genes regulated by TNF-α is the TNF-α-induced protein 6 (TNFAIP6) gene, coding for the secretory protein TSG-6, initially identified in human fibroblasts lines and peripheral blood mononuclear cells [85,86]. TSG-6 can be induced by diverse cytokines (i.e., TNF, IL-1) and other inflammatory stimuli (i.e., LPS) and is a member of the family of hyaluronate (HA) binding proteins involved in cell–cell and cell–matrix interactions during inflammation and tumorigenesis [85,87,88]. In fact, although constitutively expressed in some tissues, including the brain and spinal cord, TSG-6 is generally upregulated in response to inflammation in a wide variety of cell types, such as monocytes/macrophages, dendritic cells, astrocytes, mesenchymal stem/stromal cells (MSCs), vascular smooth muscle cells (VSMCs), and fibroblasts [89,90,91,92,93]. TSG-6 has a variety of activities, including the modulation of immune and stromal cells function and the regulation of extracellular matrix organization and interaction with cell surface receptors and soluble mediators (e.g., chemokines) (figure in Section 4) [94]. The anti-inflammatory functions of TSG-6 rely on direct modulation of inflammatory cells and on the regulation of the assembly/organization of HA matrices that underlie its immunosuppressive properties [95,96,97,98,99,100,101,102]. TSG-6 was also shown to inhibit NETs release from bone-marrow-derived neutrophils [103]. Moreover, the original evidence that TSG-6 produced by MSCs underlies their immunomodulatory and reparative functions [104,105] has stimulated interest in understanding the role of this molecule in a number of pathological conditions. Indeed, an increasing body of experimental evidence suggests that the main function of this protein is to protect tissue from the damaging effects caused by inflammation, as demonstrated in animal models of acute myocardial infarction [104], atherosclerosis [106], lung injury [107,108], arthritis [109,110,111,112,113], spinal cord injury [114], and, most notably, acute brain injury [115,116,117,118,119,120].

## 4. Immunomodulatory Functions of TSG-6 In Vitro

TSG-6 was the first soluble chemokine-binding protein to be identified in mammals, as it binds to a wide range of chemokines belonging to both CC and CXC subfamilies, hindering their presentation and interaction with matrix molecules [121,122]. By interacting with CXCL8, TSG-6 blocks its binding to endothelial heparan sulphate (HS), a mechanism implicated in its potent (in vitro and in vivo) inhibitory activity on neutrophil chemotaxis and transendothelial migration (Figure 1) [121,122,123,124,125]. The latter mechanism has been reported to underlie the protective effects of TSG-6 in a number of degenerative and inflammatory experimental conditions, including acute brain injury [104,105,117,126,127,128]. Moreover, binding to other chemokines that can be recognized by diverse leukocytes (e.g., CCR5, CCR7, and CXCR4) strongly suggests that TSG-6 may regulate recruitment and migration of different white blood cells (Figure 1) [121,122]. Accordingly, TSG-6 was reported to regulate the function of dendritic cells, macrophages, monocytes, and T cells attenuating their inflammatory responses and promoting immune tolerance [96,104,107,127,129,130,131,132,133].

In human neutrophils, TSG-6 is constitutively present in the secretory lactoferrin-positive granules; monocytes, macrophages, myeloid dendritic cells, and neutrophils produce high levels of TSG-6 in response to inflammatory triggers (LPS or cytokines) [134]. Conversely, anti-inflammatory cytokines, namely IL-4 and IL-10, dampen LPS-induced TSG-6 expression in human leukocytes [134].

In turn, TSG-6 acts in an autocrine mode on macrophages to promote their transition from proinflammatory toward anti-inflammatory phenotypes (Figure 1) [107,133,135]. In particular, this immunomodulatory protein inhibits the association between TLR-4 and MyD88, thus suppressing the activation of NF-κB, and prevents the expression of pro-inflammatory mediators (e.g., IL-6, TNF-α, IL-1β and inducible nitric oxide synthase, iNOS, STAT1, and STAT3), while elevating the expression of anti-inflammatory proteins (e.g., CD206, IL-4, and IL-10) [96,98,107,129,136,137].

The majority of the studies performed to date on TSG-6 were focused on its ability to mediate the therapeutic efficacy of MSC in a number of pathological conditions, including neuroinflammation, since it plays a major role in the modulation of sterile inflammation [94,138]. Bone MSCs stimulated with TNF-α release TSG-6 that has been reported to inhibit production of inflammatory cytokines triggered by LPS in cultured rat astrocytes [115,139]. This effect was mediated by inhibition of NF-κB signaling pathway and was suggested to underlie the protective role of bone NSCs on BBB damage induced by intracerebral or subarachnoid hemorrhage in rat [115,139].

Similarly, MSCs and adipose-derived stem cells (ADSCs) exposed to TNF-α increased their expression of TSG-6 that, once secreted, inhibited the release of inflammatory mediators (i.e., IL-1β, IL-6, TNF-α, iNOS) by cultured BV2 microglia [140,141]. This effect was shown to be mediated by the interaction of TSG-6 with CD44 and subsequent blockade of NF-κB inflammatory signaling in BV2 microglia [141]. In fact, TSG-6 binds and forms stable complexes with HA, thus stabilizing its interaction with the membrane receptor CD44, and resulting in negative regulation of TLR-4-dependent signaling [95,97,142,143]. Accordingly, TSG-6 downregulated the TLR2/MyD88/NF-κB signaling (Figure 1) and reduced the production of proinflammatory cytokines, such as IL-1β, IL-6, and TNF-α, in primary microglia treated with a specific TLR2 agonist, an effect that was suggested to underlie the ability of TSG-6 to attenuate neuropathic pain caused by chronic constriction injury in rats [144]. A similar mechanism was described in cultured murine macrophages, whereby TSG-6 released by stimulated MSCs downregulated TLR2-mediated nuclear translocation of NF-κB, by binding CD44 directly or through a complex with HA [98]. The ability of TSG-6 to inhibit proliferation and release of inflammatory mediators was also suggested to occur through reduced activation of p38 and JNK signaling in rat macrophages [145,146].

TSG-6 also suppressed LPS-induced TNF-α production and the expression of the inflammatory M1 phenotype in human monocyte-derived macrophages [147], while it prevented NF-κB activation by inhibiting the association of MyD88 with TLR4 in mouse macrophages (Figure 1) [107]. Interestingly, TLR2-related pathways were reported to play a major role in the regulation of the polarization state of microglia, while NF-κB and p38 were implicated in polarization of microglia/macrophages during ischemic stroke injury [148,149,150,151,152,153]. This strongly supports the hypothesis that TSG-6 may affect M1 vs. M2 polarization shift of myeloid cells under neuroinflammatory conditions. In line with this speculation is the evidence that BV2 microglia treated with MSCs displayed reduced LPS-induced expression of early and late markers of M1 phenotype (iNOS, IL-1β, CD16, CD86), and concomitant elevation of typical markers of M2 polarization (CD206, Arg1) [154,155]. Notably, most of these effects were lost when LPS-primed BV2 microglia were exposed to MSCs in which TSG-6 was knocked down [154]. In this context, TSG-6 was suggested to modulate microglia polarization shift by preventing LPS-induced phosphorylation of STAT3 [154]. The inhibitory effect of TSG-6 on STAT3 phosphorylation was also demonstrated to underlie polarization toward M2 phenotype in murine macrophages, thus underlying protection in mice undergone LPS-induced inflammatory lung injury or liver inflammation due to alcoholic hepatitis [107,156].

## 5. TSG-6 as Mediator of the Neuroprotective Functions of MSCs

The anti-inflammatory and immunoregulatory functions of TSG-6 have been studied in a number of neurological disorders (Figure 2), although most studies mainly focused on the role of this multifunctional protein in the protective effects of MSCs, while little is known on its direct mechanisms and on its functions when released from endogenous sources.

In global cerebral ischemia caused by cardiac arrest in rat, intravenous administration of MSCs significantly reduced brain damage by upregulating cerebral expression of TSG-6, while attenuating the ischemia-induced elevation of neutrophil elastase and of inflammatory cytokines (i.e., IL-1β, IL-6, and TNF-α) expression [119,157]. Although administration of recombinant TSG-6 in the lateral ventricle was effective in reducing histological and functional deficits associated with global ischemia, these authors did not investigate the mechanisms involved in cerebral upregulation of TSG-6 by MSCs, but they hypothesized that TSG-6 was actually released by MSCs migrating to the brain [119]. In fact, intravenous infusion of MSCs, in which TSG-6 expression was silenced by siRNA, failed to attenuate brain inflammation in ischemic rats [157].

Interestingly, bone marrow-derived MSCs are effective in ameliorating stroke outcomes and, despite limited knowledge of their exact neuroprotective mechanisms, their immune-suppressive and anti-inflammatory effects are likely to play a crucial role. This is supported by the evidence that intravenous injection of MSCs in mice reduced the elevation of brain and blood levels of the inflammatory complement component C3 induced by focal cerebral ischemia, thus resulting in reduced infarct volume [158]. These effects were also associated with elevation of brain and blood levels of TSG-6, 8 h after MSCs administration, strongly suggesting its involvement in their protective effects [158].

The ability of TSG-6 to attenuate neuroinflammatory reactions was also demonstrated in rats exposed to collagenase to induce intracerebral hemorrhage (ICH), intravenously transplanted with MSCs. In this model, the protective effects of MSCs on BBB and their anti-inflammatory effects were ascribed to the increased mRNA and protein expression of TSG-6 detected in the brain 24 h and 48 h after ICH [159]. In particular, although not directly demonstrated, TSG-6 elevation was suggested to underlie suppression of the activation of NF-kB signaling pathway and downstream reduction of iNOS and peroxynitrite levels [159]. More recent studies have confirmed the ability of bone MSCs to reduce neurological deficits and BBB damage caused by ICH or SAH in rat, through TSG-6 release, since these protective effects were abolished after silencing TSG-6 by siRNA [115,139]. In particular, Tang et al. (2021) speculated that TSG-6, likely secreted via exosomes by bone MSCs, acts through a paracrine mechanism to regulate activated astrocytes to preserve BBB integrity. In addition, TSG-6 was demonstrated to mediate the anti-inflammatory effects of bone MSCs, by inhibiting NF-κB signaling pathways and the cerebral increase of inflammatory cytokines (e.g., IL-1β, IL-6, INF-γ) and peroxynitrite caused by ICH or SAH in rats [115,139].

Enhanced expression of TSG-6 and downstream suppression of NF-κB signaling pathways have also been suggested to underlie the anti-inflammatory effects of MSCs or NS309 (a small conductance Ca^2+^-activated K^+^ channels activator) in rodent models of traumatic brain injury (TBI). In fact, intravenous MSCs transplantation 2 h after TBI, or i.p. administration of NS309 30 min before TBI, caused upregulation of TSG-6 in the injured cortex up to 72 h after the insult in rat [120,160]. This was associated with reduced levels of inflammatory cytokines (i.e., IL-1β, IL-6, IL-17, TNF-α, INF-γ) and chemokines (i.e., MCP-1, MIP-2, RANTES), lower density of inflammatory microglia/macrophages and peripheral infiltrating leukocytes, together with elevated levels of anti-inflammatory cytokines (i.e., IL-10, IL-4, TGF-β1) in the lesioned tissue [120,160]. Notably, knockdown of TSG-6 using in vivo transfection with TSG-6 specific shRNA partially reversed the protective and anti-inflammatory effects of NS309 against TBI [160]. To further support its beneficial effects, there is the evidence that intravenous treatment with TSG-6 decreased neutrophil extravasation, matrix metalloproteinase (MMP)-9 expression and the resulting BBB leakage caused by TBI in mice [117]. Remarkably, acute administration of TSG-6 within 24 h of TBI not only reduced brain lesion size at two weeks, but it also promoted neurogenesis and attenuated long-term consequences of TBI, such as memory impairments and depressive-like behavior [117].

## 6. Roles of Endogenous TSG-6 in Neurodegeneration

Although most studies have focused on TSG-6 as pivotal mediator of the beneficial effects of MSCs, there is also evidence of the role of this multifunctional glycoprotein as endogenous immunoregulatory and anti-inflammatory mediator in brain pathological conditions. TSG-6 is physiologically expressed in the developing rat brain, displaying different expression patterns in distinct cerebral regions where it may play a role in oligodendrocyte maturation and neuronal precursor cell migration [161]. Although its expression during embryonic development is still controversial, other findings have demonstrated the presence of TSG-6 in astrocytes of the mature rat brain and spinal cord [92]. In fact, TSG-6 has been implicated in astrocyte maturation, since fewer GFAP+ astrocytes were found in the brain of TSG-6 null mice [92]. Despite a selective localization of TSG-6 in astrocytes was suggested by some studies, other findings have shown a more widespread distribution of this multifunctional protein.

### 6.1. Ischemic Stroke

In the brain of stroke patients, both mRNA and protein levels of TSG-6 were elevated in the peri-infarct and infarcted tissue as compared to contralateral hemisphere, being its positive staining associated with inflammatory mononuclear cells and damaged neurons in patients surviving from 3 to 29 days after stroke [162]. This was coincident with elevation of HA levels in infarcted brain tissue and serum of patients up to 37 days after stroke; concomitantly, increased expression of the HA receptor CD44 was mainly observed in dead or dying neurons from infarct or peri-infarct tissue, as well as in inflammatory mononuclear cells 3–17 days after the initial injury [162]. Increased HA synthesis and up-regulation of CD44 in microglia, macrophages, and microvessels of the ischemic brain tissue were also observed in rodents subjected to MCAo [163,164]. The preferential synthesis of high molecular weight HA in stroke tissue, together with the elevation of TSG-6, are likely aimed at modulating inflammatory responses and at re-establishing the extracellular matrix integrity during tissue remodeling after ischemic stroke [162,163,164]. According to human findings, the expression of TSG-6 was also elevated in the cerebral cortex of rats 3 days after global cerebral ischemia due to transient cardiac arrest [119]. Although the elevation of TSG-6 has been suggested to underlie its putative beneficial functions, evidence on the very early phases after ischemic stroke injury is lacking, being a time-course analysis crucial for clarifying the exact role of this multifunctional protein.

The elevation of TSG-6 is not only restricted to the ischemic brain, since recent evidence has highlighted that noncardioembolic acute ischemic stroke patients display higher plasma TSG-6 levels than control subjects [165]. Plasma TSG-6 level positively correlated with stroke severity at admission, lesion volume, neutrophil count, neutrophil-to-lymphocyte ratio, and interleukin-8 level. In those patients, elevated TSG-6 plasma levels were independently associated with three-month poor prognosis, whereas elevated TSG-6 to IL-8 ratio predicted favorable outcome at three months [165]. This latter finding adds complexity to the interpretation of TSG-6 elevation in the periphery, highlighting that the prognostic value of circulating TSG-6 levels needs to be further evaluated. Elevation of circulating levels of TSG-6 was also observed in patients with carotid stenosis or with coronary artery disease, likely deriving from monocyte-derived macrophages, endothelial, and arterial smooth muscle cells exposed to inflammatory stimuli [147,166]. Indeed, both animal and human studies have reported expression of TSG-6 in atherosclerotic lesions, strongly suggesting an atheroprotective role of this anti-inflammatory protein in vascular lesions [106,147,167,168,169]. Intriguingly, high Alzheimer’s disease neuropathologic change (ADNC) and moderate–severe cerebral amyloid angiopathy (CAA), in the absence of concurrent pathologies (e.g., infarctions, Lewy bodies), were associated with increased TSG-6 expression and HA content in cerebral microvascular lysates [170]. In high ADNC subjects, TSG-6 was not only expressed in the vasculature, but also in neurons and astrocytes that concomitantly express HA synthase 2, strongly suggesting the involvement of HA/TSG-6 interplay in the regulation of brain pathological processes [171].

### 6.2. Other Brain Injuries

Either brain or spinal cord injury resulted in considerable upregulation of TSG-6 mRNA expression, whereby the protein is associated with the glial scar, likely playing a role in formation and stabilization of this HA-rich matrix forming an immunosuppressive environment [92,172]. Upon binding to inflammatory cells, these TSG-6 modified HA matrices modulate their responses, thus contributing to pathological inflammation [100,101]. Accordingly, elevation of endogenous TSG-6 mRNA was recently found in the lesioned hemisphere of mice subjected to penetrating brain injury (PBI), an effect that was associated with an anti-inflammatory role of this protein in the glial scar [172]. Indeed, TSG-6 null mice display a more severe inflammatory response (i.e., higher levels of NF-κB, RANTES and IL-1β, as well as higher number of CD68+ activated microglia and macrophages) and increased glial scar deposition in the injured brain as compared to littermate control mice [172].

In a rat model of SAH, both mRNA and protein levels of TSG-6 were upregulated from 12 to 72 h after injury in the brain, with a prevalent localization in Iba1 immunopositive microglia, where TSG-6 triggers an anti-inflammatory protective phenotype through pSTAT3 regulation [116]. In fact, knockdown of endogenous TSG-6 by siRNA elevated (CD86+) M1 vs. (CD163+) M2 ratio in cerebral microglia and aggravated neurological deficits 24 h after SAH. Conversely, intracerebroventricular (i.c.v.) administration of recombinant TSG-6, 1.5 h after SAH in rat, reduced microglia shift toward inflammatory phenotypes, attenuated TNF-α expression level, and upregulated IL-10 expression levels, thus resulting in reduced brain edema and improved neurological deficit [116,173].

The relevance of inflammatory triggers to induce TSG-6 expression is also highlighted by the evidence that elevation of TSG-6 gene expression was observed in the neonatal brain 4 h after intraperitoneal injection of LPS, where it was suggested to represent a compensatory anti-inflammatory mechanism [161]. In fact, treatment with recombinant TSG-6 exerted neuroprotection by reducing systemic inflammatory responses and cerebral apoptosis caused by LPS in newborn rats [161].

## 7. Conclusions

Elevation of endogenous TSG-6 has been reported to occur upon diverse acute and chronic neurodegenerative conditions, as a compensatory response aimed at triggering anti-inflammatory and a multitude of (neuro)protective mechanisms. Thus, emerging evidence provides insights into the potential use of TSG-6 as a peripheral biomarker for diagnostic and/or prognostic purposes, especially in the context of ischemic stroke, where the pathobiological relevance of this protein was also validated in patients. Moreover, TSG-6 mediates most of the beneficial functions of MSCs in acute neurodegenerative conditions, including cerebral ischemia, and is itself able to provide neuroprotection when administered to animal models. All these findings strongly highlight the promising efficacy of TSG-6 replacement therapy, with either the peptide itself or its analogues, particularly against acute neurodegenerative insults, and pose the basis for further investigation aimed at characterizing its pharmacokinetic and pharmacodynamic properties in these pathological contexts.

## Figures and Tables

**Figure 1 ijms-24-01162-f001:**
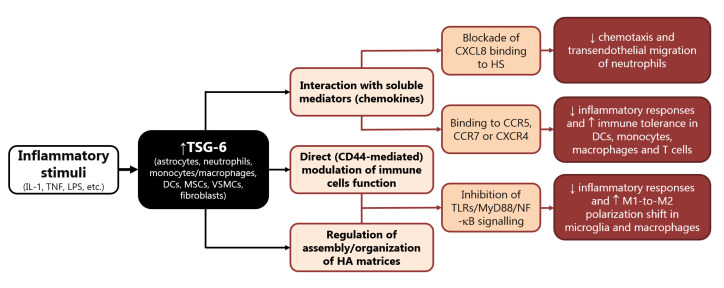
Main mechanisms involved in the immunomodulatory functions of TSG-6. Abbreviations: DC: dendritic cells, HA: hyaluronate, HS: heparan sulphate, IL: interleukin, LPS: lipopolysaccharide, MSCs: mesenchymal stem/stromal cells, myeloid differentiation primary response: MyD, TLRs: Toll-like receptors, NF-κB: nuclear factor kappa B, TNF: tumor necrosis factor, TSG: tumor necrosis factor-α-stimulated gene, VSMCs: vascular smooth muscle cells.

**Figure 2 ijms-24-01162-f002:**
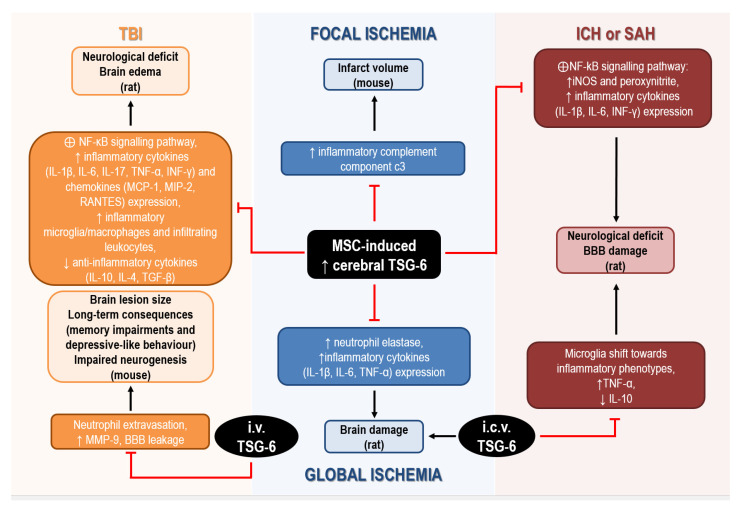
Mechanisms underlying the neuroprotective effects of TSG-6 in rodents subjected to acute brain injury. Abbreviations: BBB: blood–brain barrier, IL: interleukin, INF: interferon, MCP: monocyte chemoattractant protein, MIP: macrophage inflammatory protein, MMP: matrix metalloprotease, MSC: mesenchymal stem/stromal cells, TGF: transforming growth factor, TNF: tumor necrosis factor, TSG: tumor necrosis factor-α-stimulated gene.

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
