# Peer review of "Tumor Necrosis Factor (TNF)-α-Stimulated Gene 6 (TSG-6): A Promising Immunomodulatory Target in Acute Neurodegenerative Diseases"

_ijms, 2023, doi:10.3390/ijms24021162_

Round 1

Reviewer 1 Report

The authors present a comprehensive review of the role of TSG-6 in modulating the inflammatory response to acute neuronal injury. This is an impressive amount of work that is, for the most part, well-written. It is rather dense with biological pathways, and a few minor adjustments would make it easier to follow. My suggestion are as follows:

1.) The title is a little misleading. The review focuses, almost exclusively, on the role of TSG-6 in stroke/ischemia, with some references to its role in acute TBI. Neurodegenerative diseases, which most commonly covers AD, PD, ALS, MND, HD etc. are not covered at all, with the exception of two lines towards the end (336-342).

2.) Similarly, section 6.2 ‘other neurodegenerative conditions’, only covers acute or penetrating CNS injury, such as those that result in a glial scar, and SAH. Revision of this sub-title is recommended.

3.) Section 2 is dense with biological pathways relating to the ischemic immune response.  While comprehensive, much of this information does not directly relate to the role of TSG-6, and as such this section could be more succinct.

4.) The review would benefit from another illustrative figure, either to assist with comprehension of section 2 and/or the biological pathways associated with TSG-6.

5.) Figure 1 is a little confusing. Generally, the use of an arrow (-->) indicates that A promotes B, and the use of an inhibition symbol (--| ) indicates that A inhibits B. Currently, Figure 1 reads as (e.g) BBB leakage promotes increased neurogenesis, and this is inhibited by TSG-6.

Minor comments:

1. Lines 37-40 do not make grammatical sense, and revision is suggested.

2. Lines 48 – 50 do not make grammatical sense, and revision is suggested. 

3. A few words have also been used, that do not make sense given their current placement:

1.) line 44 – concur; 2.) line 49 – rational immodulation; 3.) line 65 – ‘being’ should come after ‘the latter’

Author Response

Answers to the comments of Reviewer #1:

The authors present a comprehensive review of the role of TSG-6 in modulating the inflammatory response to acute neuronal injury. This is an impressive amount of work that is, for the most part, well-written. It is rather dense with biological pathways, and a few minor adjustments would make it easier to follow.

R: We thank the Referee for the very accurate revision of our manuscript and for the very helpful suggestions aimed at improving its quality and impact. Below we provide point-by-point responses to each comment, including a revised version of the manuscript that has been amended according to the Reviewer's requests.

My suggestion are as follows:

1.) The title is a little misleading. The review focuses, almost exclusively, on the role of TSG-6 in stroke/ischemia, with some references to its role in acute TBI. Neurodegenerative diseases, which most commonly covers AD, PD, ALS, MND, HD etc. are not covered at all, with the exception of two lines towards the end (336-342).

R: We entirely agree with the Referee, therefore we have modified the title by specifying that the review focuses on “acute” neurodegenerative disease (which is indeed the setting where the majority of the studies on the beneficial functions of TSG-6 were performed).

2.) Similarly, section 6.2 ‘other neurodegenerative conditions’, only covers acute or penetrating CNS injury, such as those that result in a glial scar, and SAH. Revision of this sub-title is recommended.

R: We agree with this comment and changed sub-title as “Other brain injuries”

3.) Section 2 is dense with biological pathways relating to the ischemic immune response.  While comprehensive, much of this information does not directly relate to the role of TSG-6, and as such this section could be more succinct.

R: Following the Reviewer’s suggestion, we have shortened this paragraph, limiting the discussion to the mediators/mechanisms directly related to the immunoregulatory effects of TSG-6 (i.e., inflammatory/anti-inflammatory cytokines, TLRs/NFkB pathways, monocytes/neutrophils migration and polarization shift, etc.)

4.) The review would benefit from another illustrative figure, either to assist with comprehension of section 2 and/or the biological pathways associated with TSG-6.

R: As suggested, we have added a new figure (Figure 1) on the biological pathways associated with TSG-6 relevant for its immunomodulatory functions. We have added a legend and references to the figure in the text; while we have changed numbering of the previous picture (as Figure 2).

5.) Figure 1 is a little confusing. Generally, the use of an arrow (-->) indicates that A promotes B, and the use of an inhibition symbol (--| ) indicates that A inhibits B. Currently, Figure 1 reads as (e.g) BBB leakage promotes increased neurogenesis, and this is inhibited by TSG-6.

R: We thank the Reviewer for highlighting this imprecision and we have modified the figure to make it less confusing.

Minor comments:

Lines 37-40 do not make grammatical sense, and revision is suggested.

R: We have modified the sentence as follows: “These interventions are based on reperfusion of the infarcted area, while they do not exert direct neuroprotection to block the progression of brain damage that remains a hopeful target”

  1. Lines 48 – 50 do not make grammatical sense, and revision is suggested. 

R: We have modified the sentence as follows: “To implement therapeutic approaches, recent experimental work has suggested that an appropriate immunomodulation would allow to achieve neuroprotection”

  1. A few words have also been used, that do not make sense given their current placement:

1.) line 44 – concur; 2.) line 49 – rational immodulation; 3.) line 65 – ‘being’ should come after ‘the latter’

R: We have modified the words as follows: 1) “concur” was replaced with “contribute”; 2) “rational immunomodulation” was substituted with “appropriate immunomodulation”, 3) “being” was placed after “the latter”

Reviewer 2 Report

This review by La Russa et al. discusses recent data on the involvement of Tumor necrosis factor-α-stimulated gene 6 (TSG-6) in neurodegenerative disorders, with emphasis on relevant anti-inflammatory and immunomodulatory functions, especially in stroke. They show evidence supporting the potential use of TSG-6 as a peripheral biomarker for diagnostic of ischemic-related episodes, and highlight the promising efficacy of TSG-6 replacement therapy against acute neurodegenerative diseases. 

I think all the relevant literature is properly cited and the authors make a good job of presenting the evidence to make up their case. 

The only comment I have is minor. The authors may want to go over the text carefully to correct a few spelling mistakes. Nothing overtly major, grammatically this manuscript is very sound, just a few spelling mistakes. 

Author Response

We thank the Reviewer for appreciating our work and for highlighting the presence of spelling mistakes that we have carefully checked and corrected in the text.